# *TOMM40* RNA Transcription in Alzheimer’s Disease Brain and Its Implication in Mitochondrial Dysfunction

**DOI:** 10.3390/genes12060871

**Published:** 2021-06-06

**Authors:** Eun-Gyung Lee, Sunny Chen, Lesley Leong, Jessica Tulloch, Chang-En Yu

**Affiliations:** 1Geriatric Research, Education, and Clinical Center, VA Puget Sound Health Care System, Seattle, WA 98108, USA; eun-gyung.lee@va.gov (E.-G.L.); sunny.chen@va.gov (S.C.); lesley.leong@va.gov (L.L.); jessica.tulloch@va.gov (J.T.); 2Department of Medicine, University of Washington, Seattle, WA 98195, USA

**Keywords:** *TOMM40* gene, Alzheimer’s disease, RNA transcription, pseudogene, mitochondrial dysfunction

## Abstract

Increasing evidence suggests that the Translocase of Outer Mitochondria Membrane 40 (*TOMM40*) gene may contribute to the risk of Alzheimer’s disease (AD). Currently, there is no consensus as to whether *TOMM40* expression is up- or down-regulated in AD brains, hindering a clear interpretation of *TOMM40*’s role in this disease. The aim of this study was to determine if *TOMM40* RNA levels differ between AD and control brains. We applied RT-qPCR to study *TOMM40* transcription in human postmortem brain (PMB) and assessed associations of these RNA levels with genetic variants in *APOE* and *TOMM40.* We also compared *TOMM40* RNA levels with mitochondrial functions in human cell lines. Initially, we found that the human genome carries multiple *TOMM40* pseudogenes capable of producing highly homologous RNAs that can obscure precise *TOMM40* RNA measurements. To circumvent this obstacle, we developed a novel RNA expression assay targeting the primary transcript of *TOMM40*. Using this assay, we showed that *TOMM40* RNA was upregulated in AD PMB. Additionally, elevated *TOMM40* RNA levels were associated with decreases in mitochondrial DNA copy number and mitochondrial membrane potential in oxidative stress-challenged cells. Overall, differential transcription of *TOMM40* RNA in the brain is associated with AD and could be an indicator of mitochondrial dysfunction.

## 1. Introduction

Understanding the role that genetics plays in the pathogenesis of AD has been a major research focus for the past three decades. These collective efforts have provided valuable insights into the molecular mechanisms associated with this disease. The advancement of genome-wide approaches has led to the identification of more than 40 AD-associated genetic loci. However, most of these loci have only moderate effect sizes with odds ratios ranging from 1.1 to 1.5 (AlzGene), except for the apolipoprotein E gene (*APOE*) which has an odds ratio of 3.7. The strength of the association between *APOE* and AD risk is orders of magnitude larger than all other AD loci combined, suggesting that this locus is a major biological contributor to the risk of AD. Therefore, deciphering the mechanistic role of the *APOE* locus in AD should provide insight into the etiology of this devastating disease.

Besides *APOE* itself, the extended region surrounding *APOE* has also been consistently identified by genome-wide association studies (GWAS) to strongly associate with AD [1,2,3,4]. This extended region consists of at least three additional genes (i.e., *NECTIN2, TOMM40,* and Apolipoprotein C1 (*APOC1*)), which carry out specific cellular functions that may possibly intersect with AD pathophysiology. Because of the strong linkage disequilibrium (LD) of these genes with *APOE*, researchers have always assumed that the disease-associated genetic signals from these genes solely reflect their associations with the *APOE* ε4 allele. However, increasing evidence points to a different interpretation. For example, a genome-wide linkage study of 71 Swedish late onset AD families found that the strongest signal in a multipoint linkage analysis of *APOE* ε4-negative families still resided in the *APOE* region [5]. Furthermore, multiple studies have shown that individuals who carry an African ε4 haplotype of *APOE* have less risk of developing AD when compared to those with a Caucasian ε4 haplotype [6,7,8]. These observations suggest the presence of loci in this region, beyond *APOE*, that may influence AD risk. One strong candidate is the *TOMM40* gene.

*TOMM40* encodes a mitochondrial channel protein TOM40, which is essential for the formation of a translocase of the mitochondrial outer membrane (TOM) complex [9]. The TOM complex is involved in the recognition and import of nuclear-encoded proteins into the mitochondria [10]. Alterations of mitochondrial metabolism have gradually been accepted as prominent features in AD and mitochondrial dysfunction is a known characteristic of the disease [11,12,13,14]. Mitochondrial degeneration has shown to be an early sign of AD pathology, appearing even before neurofibrillary tangles (NFT) [15]. Damages to both the components and structure of mitochondria are extensively reported in AD [16], and the deficiency of several key antioxidant enzymes is a well-established hallmark of the AD brain [17]. Thus, abnormal mitochondrial dynamics, including components, morphology, membrane potential, and DNA copy number could contribute to AD risk [15,18].

*TOMM40* has not only been genetically linked to AD risk but may also be functionally connected with AD pathophysiology. In a Chinese cohort, SNPs in *TOMM40* remained statistically significantly associated with AD after adjusting for age, sex, and *APOE* ε4 status [19]. A deoxythymidine homopolymer (poly-T) at rs10524523 within intron 6 of the *TOMM40* has been associated with the risk and age at onset of AD [20,21,22,23]; and the “VL” variant of this poly-T marker has been associated with increased mRNA expression of both *TOMM40* and *APOE* in *APOE* ε3/ε3 brain [24]. In addition, our own high-density SNP association studies identified genetic variants in *TOMM40* to be strongly associated with AD in Caucasians, after controlling for the *APOE* ε2/ε3/ε4 alleles [25,26]. Our quantitative trait loci studies showed that there is an association between *TOMM40* SNPs and apoE protein levels in both cerebrospinal fluid and PMB, suggesting that genetic variation within *TOMM40* may be associated with *APOE* and *TOMM40* expression in the human brain [27,28,29]. Furthermore, there is evidence supporting a direct connection between *TOMM40* and Aβ activity. For example, the Aβ peptide is imported into the mitochondria via the TOM40 protein [30] and the amyloid precursor protein has been reported to be associated with TOM40 in AD, but not controls [31]; Aβ peptides and mis-directed amyloid precursor protein interfere with mitochondrial protein import and disrupt mitochondrial function [31,32,33]; and the accumulation of Aβ in mitochondria leads to the overproduction of reactive oxygen species [30,34]. Given that *TOMM40* appears to be involved in APP/Aβ translocation and metabolism as well as *APOE* regulation, it is plausible that *TOMM40* plays a role in AD via effects on mitochondrial function. Consequently, *TOMM40* expression levels may be impacting mitochondrial function and contributing to AD risk.

Expression of *TOMM40* has been investigated in peripheral blood. Numerous studies consistently showed lower *TOMM40* mRNA levels in AD blood samples compared to controls [35,36,37,38], and a decrease in TOM40 protein level has also been observed in AD blood [37]. However, studies using human PMB are scarce and have generated conflicting results. For example, *TOMM40* mRNA levels were reported as both increased and decreased in the AD frontal cortex [39], or significantly increased in AD temporal and occipital cortices [24]. Currently, there is no consensus as to whether *TOMM40* gene expression is up- or down-regulated in AD brains, and this inconsistency hinders a unified clear interpretation of *TOMM40*’s role in AD risk. The aim of this study was to definitively determine if PMB *TOMM40* mRNA levels differ between AD and control subjects.

## 2. Materials and Methods

### 2.1. Human PMB and Cell Lines

This work used deidentified human biospecimens that have already been collected by other established programs. Therefore, no consent was obtained for this work. Previously, all human specimens were obtained from the University of Washington (UW) Alzheimer’s Disease Research Center after approval by the institutional review board of the Veterans Affairs Puget Sound Health Care System (MIRB# 00331). AD patient diagnosis was confirmed postmortem by neuropathological analysis. Clinically normal subjects were volunteers who were over 65 years of age, never diagnosed with AD, and lacked AD neuropathology at autopsy. AD Brains exhibited Braak stages between V and VI, whereas control brains exhibited Braak stages between I and III. Postmortem frontal lobe tissues were obtained from the middle frontal gyrus tissues that had been rapidly frozen at autopsy (<10 h after death) and stored at −80 °C until use. Demographics of the study sample are listed in Table 1. Hepatocytoma HepG2, glioblastoma U-87 MG and U-118 MG cells (ATCC) were grown in 89% Dulbecco’s modified Eagle’s medium (DMEM) (Gibco); neuroblastoma SH-SY5Y cells (ATCC) were grown in 89% DMEM with F12 (Gibco). Both media were supplemented with 10% fetal bovine serum (FBS) (Gibco). Glioblastoma LN-229 cells (ATCC) were grown in 94% DMEM supplemented with 5% FBS. All cell cultures were supplemented with 1% penicillin/ streptomycin (Invitrogen) and cultured at 37 °C in a 5% CO_2_ atmosphere.

### 2.2. DNA/RNA Extraction and Genotyping

Genomic DNA and RNA were isolated from frozen PMB using the AllPrep DNA/RNA Mini Kit (Qiagen). Nucleic acid concentrations were measured by NanoPhotometer (Implen), and samples were stored at −20 °C prior to use. SNPs (assay #) were genotyped using TaqMan allelic discrimination assays purchased from Thermo Fisher Scientific as follows: rs429358 (C_3084793_20), rs7412 (C_904973_10), rs71352238 (C_98078714_10), rs2075650 (C_3084828_20), rs741780 (C_3084816_10), and rs10119 (C_8711595_10). All procedures were performed according to the manufacturers’ protocols. For rs10524523 (poly-T) S/L/VL typing, PCR was performed using primers chr19_50094846F (5′-cctccaaagcattgggatta) and chr19_50095058R (5′-gggacagggaaagaaaacaa). The length of the amplicons was then determined using a QIAxcel Advanced system (Qiagen) based on a high-resolution capillary electrophoresis. The expected amplicon size was calculated to be 179 bp + poly-T length in bp. The observed amplicon size was ≤198 bp for the S variant (poly-T ≤ 19); 199–208 bp for the L variant (poly-T = 20–29); and ≥ 209 bp for the VL variant (poly-T ≥ 30).

### 2.3. Sequence Alignment and Phylogenetic Tree

The *TOMM40* mRNA and *TOMM40*L transcript reference sequences were obtained from NCBI Nucleotide database. All *TOMM40* pseudogene sequences were extracted from UCSC Genome Browser’s UCSC DAS server, using the genomic coordinates (version hg38) obtained from NCBI Genes & Expression’s Gene database. The nucleotide sequences were aligned using NIH’s BLAST blastn program. The query was optimized for highly similar sequences (megablast). The extracted *TOMM40* pseudogene sequences were blasted against *TOMM40* mRNA reference sequence to query for percent identity. The Phylogenetic Tree was generated from Molecular Data with MEGA (https://doi.org/10.1093.molbev/mst012, accessed on 22 March 2021). The bootstrap value (or node) was calculated from resampling analysis as an indicator of good confidence in specific node. The substitution rate is defined as the number of nucleotides that were substituted per site per unit time.

### 2.4. Conventional End-Point PCR and Gel Electrophoresis

Expression of pseudogene RNAs was examined by end-point PCR. Total RNAs were extracted from cells and cDNA synthesis was performed using the PrimeScript RT Reagent Kit (Takara Bio, Mountain View, CA, USA). Pseudogene-specific primer sets were used to amplify each pseudogene template or cDNA. Information on the pseudogene-specific primers is listed in Appendix A. DNA fragment analysis of the amplification reactions was performed in a QIAxcel (Qiagen).

### 2.5. Reverse Transcriptase (RT) Reaction and Quantitative PCR (qPCR) Assay

RT-qPCR assays were performed as previously reported [40]. Briefly, a fixed reverse-transcribed cDNA input (5 ng) was amplified using TaqMan assays or SYBR PCR assays in a QuantStudio 5 (Applied Biosystems, Thermo Fisher). The thermal cycling profile consisted of 2 min at 50 °C, 10 min at 95 °C, and then 40 cycles of 15 s at 95 °C and 1 min at 60 °C. The amplification efficiency of both TaqMan and SYBR PCR assays were measured by a standard curve method using serial dilutions in qPCR reactions and calculated using [10^(−1/slope)^] -1. The calculated amplification efficiency is as follows: 0.92 (total *TOMM40* mRNA); 0.90 (*TOMM40* IVS9); 0.89 (pseudogene *P1b/P2*); 0.91 (total *TOMM40* Ex4-Ex5). For each sample, qPCR assays were performed in triplicate. Information on primers, probes, and TaqMan assays is listed in Appendix A. For *TOMM40* RNA quantification, all reactions were quantified by using a fixed threshold (0.15) in the linear range of amplification and recording the number of cycles (cycle threshold, C_T_) required for the fluorescence signal to cross the threshold. To control for the quantity of input RNA, we quantified *ACTB* mRNA as an internal control for each sample and obtained a normalized ΔC_T_ value: mean of C_T_ triplicate (target)–mean of the *ACTB* C_T_ triplicate. In this setting, smaller ΔC_T_ values indicate higher RNA transcription levels. Additionally, fold change (FC) of *TOMM40* transcription levels of AD to Control subjects was computed as FC (AD) = 2^−ΔΔCt^, where ΔΔC_T_ = mean ΔC_T_ (AD)– mean ΔC_T_ (Control) [41].

### 2.6. Fraction Estimation of Pseudogene RNA and Surrogate RNA Using Digital PCR (dPCR)

The *P1b/P2* primer set was used to measure levels of pseudogene RNAs and IVS9 primers were used to amplify *TOMM40* surrogate RNA by qPCR. A primer set spanning Ex4 and Ex5 was used for measuring levels of the total *TOMM40* RNA pool. These primers were also used for RT-qPCR (SYBR) assays as listed in Appendix A. We performed absolute quantification of RNA levels by QIAcuity dPCR (Qiagen). The QIAcuity carries out fully automated processing including all necessary steps of plate priming, sealing of partitions, thermocycling, and image analysis. We used the the QIAcuity Nanoplate 26K 24-well. For each well, 40 μL reaction contained 13.3 μL of 3x EvaGreen PCR master mix (Qiagen), 0.4 μM of each forward and reverse primer, and a fixed concentration of cDNA template 3 μL (15 ng). The thermal cycling program consisted of 2 min at 95 °C, 40 cycles of 15 s at 95 °C, 20 s at 55 °C, and 1 min at 72 °C, and then 5 min at 40 °C. We computed the fraction of target RNA (pseudogene RNA or surrogate IVS9 RNA) by dividing the number of copies/μL of target RNA by the number of copies/μL of total *TOMM40* RNA pool. For the quality control of QIAcuity dPCR, we replicated the assay with different amounts of template input and showed the reproducibility of the fraction of the target RNA in the total *TOMM40* RNA pool.

### 2.7. Hydorogen Peroxide Treatment

Twenty-four hours prior to treatment, the cells were seeded at a density of 70–80%. For RNA transcription and mitochondrial DNA (MtDNA) copy number assays, cells were seeded on a 6-well plate, whereas a 96-well plate was used for the mitochondrial membrane potential assay. We searched the literature for the effects of hydrogen peroxide on mitochondrial function. Based on previously published conditions, we tested multiple concentrations (100 µM, 200 µM, 250 µM, 500 µM and 1 mM) of hydrogen peroxide in the cell lines and selected 500 µM as an optimal concentration that maintained good cell viability and had noticeable effects on mitochondrial function. The seeded cells were then treated with 500 μM Hydrogen peroxide, H_2_O_2_, (Sigma) in growth media. For controls, the same number of cells were plated and cultured without H_2_O_2_. Cells were collected 24 h post-treatment, subjected to genomic DNA and total RNA isolation, followed by measurement of MtDNA copy number and RNA transcription levels. Three to four independent treatments with H_2_O_2_ were performed.

### 2.8. MtDNA Copy Number Assay

Reactions for MtDNA copy number count and single copy reference gene, *HGB* (Hemoglobin), were run separately with 10 ng of DNA in a 384-well optical plate. Each reaction was run in triplicate on QuantStudio 5 (Applied Biosystems, Thermo Fisher, Foster City, CA, USA). The 10 μL reaction included 10 ng of DNA, 5 μL of 2x Power SYBR Green PCR Master Mix (Applied Biosystems, Thermo Fisher), and 0.05 μM of each forward and reverse primer. Thermal cycling profile consisted of 2 min at 50 °C, 10 min at 95 °C, and then 40 cycles of 15 s at 95 °C, 60 s at 56 °C, and 60 s at 72 °C. The ΔC_T_ method was used to control for the quantity of input DNA for each sample by quantification of *HGB* DNA. The normalized ΔC_T_ value was calculated: mean of MtDNA C_T_ triplicate– mean of the *HGB* C_T_ triplicate. The fold change (FC) of the MtDNA copy number isolated from H_2_O_2_-treated cells to untreated cells was computed as FC (treated) = 2^-ΔΔC^_T_, where ΔΔC_T_ = ΔC_T_ (treated) − ΔC_T_ (untreated) [41].

### 2.9. Mitochondrial Membrane Potential (MMP) Assay

MMP of the human cell lines was analyzed using a MitoProbe JC-1 assay kit (Thermo Fisher). The cationic dye, JC-1 (5′,6,6′-tetrachloro-1,1′,3,3′-tetraethylbenzimidazolyl-carbocyanine iodide), exhibits potential-dependent accumulation in mitochondria, which is indicated by a fluorescence emission shift from monomeric green (529 nm) to JC-1 aggregates red (590 nm). Consequently, MMP change in response to cellular stimuli is represented by the ratio of red to green fluorescence intensity. The membrane potential disrupter, CCCP (carbonyl cyanide 3-cholorophenylhydrazone), was included in all assays as a control to confirm that the JC-1 response is sensitive to changes in membrane potential. Twenty-four hours prior to the hydrogen peroxide treatment, cells were seeded at a density of 70–80% on a black 96-well plate with a clear bottom. The seeded cells were treated with 500 μM H_2_O_2_ for 24 h and then assayed for MMP measurements. A quantity of 2 μM JC-1 was added and incubated at 37 °C, 5% CO_2_ for 30 min. The reaction plate was washed with PBS and the fluorescence was measured with 488 nm excitation and green (529 nm) or red (590 nm) emission using SpectraMax M2 plate reader (Molecular Devices). All procedures were performed according to the manufacturers’ protocols. Fold change (FC) of membrane potential of H_2_O_2_-treated cells to untreated cells was computed as FC (treated) = ratio of red to green (treated)/ratio of red to green (untreated). Three independent MMP assays were performed for each H_2_O_2_ treatment for each cell type.

### 2.10. Statistical Analyses

The qPCR data is expressed as normalized ΔC_T_ values and was calculated as follows, ΔC_T_ value = mean of C_T_ triplicate (target)–mean of the ACTB C_T_ triplicate. Statistical analyses were performed using independent samples *t*-test using the Statistical Package for the Social Sciences (SPSS) version 19 (SPSS). The MtDNA copy number assay and membrane potential assay were not performed in the same cell setting because the membrane potential assay needs to be conducted on live cells. For this reason, data were not statistically compared.

## 3. Results

### 3.1. Presence of Pseudogene RNAs Obscures Accurate Measurement of TOMM40 mRNA

The transcription levels of *TOMM40* mRNA in human brain and its association with AD risk have not been fully established. Studies have reported conflicting results of either up- or down-regulated *TOMM40* mRNA in AD brains. The measurement of a gene’s mRNA levels should be a straightforward procedure unless other complications confound this measurement; apparently, an unknown barrier exists in the measurement of *TOMM40* mRNA. To expose this obstacle, we first inspected the specificity of *TOMM40* cDNA by aligning this sequence to the human genome (hg38) using the Blat tool in the UCSC genome browser (http://genome.ucsc.edu/, accessed on 18 March 2021). Besides *TOMM40* itself, the alignment showed six additional hits, including one known gene (*TOMM40L*) with moderate homology to *TOMM40* and five loci with a high degree of sequence homology. All five of these highly homologous loci contain a *TOMM40* cDNA-like sequence lacking either *TOMM40* introns or a full open reading frame, classic characteristics of a pseudogene. When queried the public databases, we found established pseudogene records for four of the five loci in the HUGO gene database (https://www.genenames.org/, accessed on 18 March 2021) with the designated nomenclatures of *TOMM40P1, P2, P3*, and *P4*. We designated the unnamed pseudogene “*TOMM40P1b*” due to its proximity to *TOMM40P1*. The genomic location, span, and similarity to *TOMM40* cDNA of these pseudogenes are listed in Table 2. A phylogenetic tree analysis indicated that they are indeed closely related to each other (Figure 1A).

Because a large portion of the human genome’s non-coding regions, including pseudogenes, can produce RNA transcripts [42], we wondered whether these *TOMM40* pseudogenes can be transcribed into RNA. To address this question, we developed new RT-PCR assays designed to specifically amplify each pseudogene’s putative RNA transcripts and generated pseudogene-specific DNA templates to serve as positive controls. To generate pseudogene templates, we PCR amplified each pseudogene’s genomic region using primer sets (Appendix A) flanking immediate up- and down-stream segments of each pseudogene, and then purified these PCR fragments using agarose gel electrophoresis. Each pseudogene template contains the entire pseudogene sequence, thus mimicking its putative cDNA. Except for *TOMM40P1*, which is flanked by heavy repetitive sequences and could not be amplified, we successfully generated DNA templates for the remaining four pseudogenes (*P1b, P2, P3*, and *P4*). Due to high homology between *P1b* and *P2*, as seen in the phylogenetic tree (Figure 1A), we combined these two into a single template of *P1b*/*P2*. To design specific primers for the *TOMM40* pseudogene RNA assays, we first identified nucleotide variants among sequences of *TOMM40* cDNA and pseudogenes using Clustal Omega’s sequence alignment (https://www.ebi.ac.uk/Tools/msa/clustalo/, accessed on 10 June 2019). We then designed allele-specific PCR primers that carry the unique nucleotide(s) of each pseudogene at the 3′-end of the primers (Appendix A). We tested these primers’ specificities in our collection of pseudogene templates using conventional end-point PCR and capillary gel electrophoresis. The results showed a robust amplification by each specific primer set with its own template, but they also showed various degrees of cross-amplifications with other pseudogene templates (Appendix A), implying that the allele-specific primers cannot fully differentiate each pseudogene. Nevertheless, these primers provide a molecular tool that makes detection of RNA transcripts of the *TOMM40* pseudogenes feasible.

Using these pseudogene-specific primers, we tested whether putative RNAs of the *TOMM40* pseudogenes could be detected in human cell lines (HepG2, U-87, and SH-SY5Y). Because the expected amplicons of pseudogene RNAs can also be generated from pseudogenes’ corresponding genomic DNA, we performed DNase digestion for all RNA isolations and included RT-negative controls for all experiments. In these experiments, we also integrated a *TOMM40* cDNA control that was RT-PCR generated using *TOMM40* cDNA-specific primers. We then performed RT-PCR reactions using total RNA isolated from these cells. The capillary gel electrophoresis showed that all pseudogenes-specific amplicons were amplified in all cell lines tested, with no amplification from RT-negative controls (Figure 1B). None of the pseudogene primer sets amplified *TOMM40* cDNA except the primer set of *P1* (Figure 1B, lane 2), which is likely due to the high homology between *TOMM40* cDNA and *P1*. The similarity of these two sequences in the phylogenetic tree further supported this notion (Figure 1A). Together, these results indicated that all *TOMM40* pseudogenes can produce RNA transcripts and some of these transcripts closely resemble *TOMM40* mRNA.

We also attempted to generate *TOMM40* cDNA-specific primers that could separate *TOMM40* mRNA from the pseudogene RNAs. The *TOMM40* gene consists of 10 exons and six mRNA transcripts (Figure 2A,B). We applied the same allele-specific primer design method to integrate *TOMM40* cDNA-specific nucleotide variants at the 3′-end of the primers. Using this approach, we generated two *TOMM40* cDNA assays that amplify across the splicing junctions of exons 1–2 (Ex1-Ex2) and exons 3-4 (Ex3-Ex4) (Figure 2C). When subjected to RT-PCR experiments, these two assays effectively amplified cDNA templates of *TOMM40*; however, they also showed leaky attributes and cross-amplification of all pseudogene templates (Appendix A). This result suggests that RNA transcripts of *TOMM40* and pseudogenes cannot be fully separated even with the meticulously designed allele-specific primers.

Learning from these observations, we suspected that the RNA transcription levels determined by commonly used *TOMM40* gene expression assays likely include both RNA species of *TOMM40* and its pseudogenes. We then designed experiments to further examine this possibility. Because exact primer and probe sequences of the commercial assays are not publicly available, we generated a *TOMM40* cDNA assay that spans splice junctions of exons 4-5 (Ex4-Ex5), the general area targeted by a popular *TOMM40* TaqMan expression assay (Thermo Fisher, assay Hs01587378_mH). When tested using end-point PCR, this assay cross-amplified all pseudogene templates (Figure 3A). This result is further evidence that conventional RT-PCR-based *TOMM40* assays likely quantify transcription levels of the entire *TOMM40*-related RNA pool, which consists of both *TOMM40* mRNA and its pseudogenes’ RNAs. Such assays cannot provide accurate measurements of true *TOMM40* mRNA.

### 3.2. Development of TOMM40-Specific RT-PCR Assays

To obtain authentic transcription levels of *TOMM40* mRNA, one could apply a deduction method in which pseudogene transcription levels are subtracted from the total *TOMM40*-related RNA pool. However, we found that most pseudogene-specific primer sets could cross-amplify templates of other pseudogenes (Appendix A), which made the precise measurement of each pseudogene RNA level unfeasible. Instead, we were only able to estimate the fraction of RNA representing pseudogenes within the total *TOMM40* RNA pool. We reasoned that the *P1b/P2* primer set can cross-amplify *P3* and *P4* in addition to its own template (Appendix A); using this one primer set allows us to access the transcription levels representing a large portion of the pseudogene RNAs while excluding amplification of the true *TOMM40* RNA. When they were compared with the results of the Ex4-Ex5 assay, which represents a measurement of the total *TOMM40*-related RNA pool, we were able to approximate the proportion of the pseudogene RNAs. We applied this strategy on total RNA isolated from a subset of PMB tissues (AD *n* = 29 and control *n* = 16) and quantified the cDNA targets using digital PCR (dPCR). From each subject, two amplicons (*P1b/P2* and Ex4–Ex5) were quantified separately, and the absolute quantification count of *P1b/P2* was then divided by the count of Ex4–Ex5 to generate the fraction. The result showed that *P1b/P2* RNA constituted around 10–18% of the total *TOMM40*-related RNA pool (Appendix A). A detailed procedure for this comparison and calculation is listed in the methods section. Provided that this estimation did not include *P1* levels, the actual fraction of the pseudogene RNAs in the total *TOMM40*-related RNA pool could be substantially higher.

The results described above prompted us to conclude that in order to eliminate cross-amplification of *TOMM40* pseudogene RNAs and accurately quantify *TOMM40* transcription, there is a need to develop an unconventional RT-PCR assay. Accordingly, we explored assays designed to target the primary RNA transcript of the *TOMM40*. We reasoned that the main difference between *TOMM40* mRNA and pseudogene RNAs lies in its primary transcript―with pseudogenes lacking intronic sequences. While the primary transcript (pre-mRNA) transcription level is expected to be only a fraction of the spliced mRNA transcription level, this proportion is likely retained consistently across samples obtained from same cell/tissue-types. Thus, an RT-PCR assay based on this principle could provide an accurate surrogate measurement of the actual mRNA transcription level, which can then be used to compare samples from human subjects.

Based on this rationale, we first inspected the RNA structure of *TOMM40* using the Ensembl RNA track in the UCSC genome browser. This track shows the presence of six variants of *TOMM40* RNA transcripts (Figure 2B). To cover the majority of these transcripts, we designed two sets of TaqMan-based assays with one extending from exon 6 into intron 6 (Ex6-IVS6) and a second one extending within intron 9 (IVS9). Map locations of these assays are shown in Figure 2C, and corresponding primers and probe sequences are shown in Appendix A. Initial conventional end-point PCR tests with their respective primers showed that both assays amplified RNA samples isolated from all four human cell lines (HepG2, U-118, U-87, and SH-SY5Y) with the expected amplicons. More importantly, no pseudogene amplicons were amplified by these new assays (Figure 3B). We next evaluated these assays in PMB tissues using a TaqMan (primers plus probe) setting in RT-qPCR. When these two assays were compared side by side, they both showed consistent expression patterns (≈2 ΔC_T_ difference) between AD and control frontal lobes (Figure 4A). Between these two assays, the IVS9 assay showed a higher sensitivity, as indicated by its lower ΔC_T_ value; thus, we selected this assay for our surrogate quantification of *TOMM40* mRNA. We estimated that RNA levels measured by the IVS9 surrogate assay represent approximately 7–20% of the total *TOMM40*-related RNA pool levels using the same dPCR approach mentioned above (Appendix A). We then applied this IVS9 assay to quantify *TOMM40* mRNA transcription levels in human PMB samples and compared these levels to the ones generated from the commercial TaqMan assay (Thermo Fisher, Hs01587378_mH). No differences in *TOMM40* mRNA transcription levels were observed between AD and control when the commercial assay was used. On the contrary, *TOMM40* mRNA showed significantly (*p* < 0.001, *t*-test) higher expression (≈2.5-fold) in AD compared to control when the IVS9 assay was used (Figure 4B).

### 3.3. Effects of TOMM40 RNA Transcription Levels

With biologically meaningful RNA measurements in hand, we further examined the relationship between *TOMM40* RNA levels and some AD-associated genotypes in human PMB. We first analyzed a set of genetic variants, including the *APOE* ε4-determing SNP rs429358 and five SNPs (rs71352238, rs2075650, rs10524523, rs741780, and rs10119) scattered across *TOMM40* (Figure 2A). For rs10524523, we stratified the “S” and “VL” variants only, excluding the third variant “L”, which is linked specifically with the ε4 variant of rs429358 that was analyzed separately. We then tested for associations between stratified alleles and *TOMM40* RNA levels quantified using our IVS9 assay. We observed significant allelic differences in rs10524523 (*p* < 0.02, *t*-test) and in rs741780 (*p* < 0.01, *t*-test; Figure 5). None of the other four SNPs showed any expression associations with their alleles. Additionally, we performed ex vivo experiments to examine the relationship between *TOMM40* RNA levels and selected mitochondria functions. We induced oxidative stress using H_2_O_2_ in human cell lines (HepG2, U-118, U-87, and LN-229) and compared *TOMM40* RNA levels, MtDNA copy number, and mitochondrial membrane potential. After H_2_O_2_ treatment, *TOMM40* surrogate RNA levels were increased (≈1.2–1.5-fold higher than the untreated one, which corresponded with a decreased DNA copy number (≈20–60% of the untreated levels) and a decreased membrane potential (≈20–75% of the untreated one; Figure 6). These findings indicate that upregulation of *TOMM40* RNA levels corresponds with mitochondrial dysfunction.

## 4. Discussion

Brain mitochondrial function plays a crucial role in neural plasticity and cognition [43] and is vital to many neural activities. Mitochondrial dysfunction occurs in a variety of psychiatric and neurodegenerative disorders [44], and is a fundamental characteristic of AD [11,12,13,14]. The *TOMM40* gene encodes a mitochondrial outer membrane translocase, which plays important roles in importing and sorting proteins for sub-mitochondrial locations. *TOMM40* is an essential gene for mitochondrial maintenance, making it a plausible candidate for influencing AD risk via mitochondrial dysfunction.

The possibility that *TOMM40* plays a direct role in AD risk has always been overshadowed by *APOE*. Located 2.1 kb upstream of *APOE*, the genetic effects of these two genes cannot easily be separated due to the strong LD structure between them [25,45]. Genetic associated signals of *TOMM40* in AD have traditionally been dismissed as surrogate signals of *APOE* [46,47]; however, this viewpoint has gradually shifted to consider *TOMM40* as an independent contributor to AD risk and healthy aging. For example, genetic variants in *TOMM40* have been consistently linked to longevity and healthy aging [48,49,50,51]. The SNP rs2075650 located in intron 2 of *TOMM40* has been considered a proxy of the SNP rs429358 that defines the ε4 allele of *APOE* [29]. The G allele of rs2075650 has been associated with a range of phenotypes including reduced longevity [52], reduced BMI [53], and increased low-density lipoprotein cholesterol [54,55], as well as an increased risk of AD [56]. Evidence also suggested that the length of rs10524523 (poly-T) within intron 6 of the *TOMM40* is linked to different levels of risk and age of onset of cognitive decline [57]. Such epidemiological data strongly support the idea that *TOMM40* plays a direct role in cognition and healthy aging. Because age is the most important risk factor for AD, the combined biological consequences of *TOMM40* and *APOE* may represent a molecular mechanism explaining the *APOE* locus’ strong genetic association with AD. Additionally, the most recent human genome project has revealed that a large number of functional sites in the genome are *cis*-regulated in nature. Thus, the unique genomic arrangement of the *TOMM40-APOE* gene cluster raises a possibility that genetic variants of the *APOE* locus could relate indirectly to mitochondrial function through LD with *TOMM40*. This concept is also in line with current trending research on the co-regulation of local genes in gene expression through the topological associating domain or 3D genome [58].

When the expression profile of a gene is altered in a disease, it provides credible evidence supporting a direct connection between that gene and the disease. *TOMM40* overexpression at both the transcriptional and translational levels in ovarian cancer has been shown to correlate with increased cell proliferation, migration, and invasion [59,60]. TOM40 protein levels are significantly reduced in brains of Parkinson’s disease patients and in α-Syn transgenic mice [61,62,63]. Significant changes were observed in the mRNA levels of mitochondrial dynamic genes such as fission/fusion-related genes and mitophagy-related genes in blood samples of AD patients [39,64]. However, the altered expression profiles (both mRNA and protein) of *TOMM40* in AD have not been clearly established. Whether the expression level of *TOMM40* is up- or down-regulated in AD brains has not been consistently observed. In one study, *TOMM40* mRNA was shown to be downregulated in 6 of 14 AD frontal lobes but upregulated in the remaining eight [39]. Such conflicting results suggest that the quantification of *TOMM40* mRNA may not be straightforward and may be complicated by other biological processes.

To determine the source of inconsistent *TOMM40* transcription levels observed in the human brain, we revisited the fundamental basics by examining the specificity of the *TOMM40* cDNA sequence. We were surprised to find that human genome carries five *TOMM40*-related pseudogenes, which all share high homology (87–96%) with *TOMM40* cDNA. We were even more surprised to find that all five pseudogenes produce RNA transcripts in various human cell lines and PMB tissues. Currently, the biological functions and/or consequences of these *TOMM40* pseudogene RNAs are undefined. The *TOMM40* pseudogenes are scattered across the human genome with unique flanking DNA sequences; thus, each of them is likely independently regulated. It is plausible that some of these RNAs can serve as templates to produce small peptides, but, to our knowledge, this has never been investigated. Another potential function of these *TOMM40* pseudogene RNAs is to regulate the transcription of *TOMM40*. Carrying highly similar sequences, these RNAs could compete with *TOMM40* mRNA for binding to proteins in transcription or post-transcription machineries. Potentially, these pseudogene RNAs could provide an RNA buffer in response to various stress/stimuli, and this buffer effect might minimize severe fluctuations in *TOMM40* RNA production. Indirect evidence supporting this concept comes from our experiment of comparing *TOMM40* cDNA assays. The commercial *TOMM40* assay, which measures both *TOMM40* mRNA and pseudogene RNAs, has much tighter transcription levels across the PMB samples when compared to the IVS9 assay that specifically measures *TOMM40* RNA only. One explanation is that this commercial assay congests all the *TOMM40*-related RNAs together, which provide a cushion effect to reduce the variability of single RNA measurement. The estimated fraction of pseudogene RNAs in the total *TOMM40*-related RNA pool is approximately 10–18% using a single pseudogene assay (*P1b/P2*) that cross-amplified four out of five pseudogenes. If the transcription levels of all five pseudogenes can be precisely measured, the overall fraction is likely to be significantly higher than this estimation. These results raise an interesting question: why does *TOMM40* gene expression need to be rigorously regulated or guarded by such a complex system from a whole genome setting?

Due to the cross-amplification of pseudogene RNAs, the conventional RT-PCR assays cannot provide an accurate measurement of *TOMM40* mRNA. This challenge prompted us to develop an alternative approach, which targets the primary RNA transcript of *TOMM40* and eliminates undesired co-measurement of pseudogene RNAs. The major difference between the primary transcript and mRNA lies in the RNA splicing. The splicing efficiency depends on the splicing kinetics, transcriptional and splicing regulators, transcription rate, intron length, exon position, RNA structure, and chromatin signatures, including histone marks and DNA methylation [65,66]. It has been shown that the splicing efficiency of pre-mRNA varies greatly across genes [67,68]. Due to the splicing process, actual transcription levels are not the same between the primary transcript and spliced mRNA. However, the level of primary transcript can provide a surrogate measurement for mRNA. Surrogate *TOMM40* transcription levels, which were measured by the primary transcript-targeted assays (Ex6-IVS6 and IVS9), had a similar profile with ≈2 ΔC_T_ value separating AD and control PMB samples. This result indicated that both measurements were consistent across samples and were suitable to serve as surrogate measurements of *TOMM40* mRNA. Between the two assays, the IVS9 has higher transcription levels (lower ΔC_T_ value) when compared to Ex6-IVS6. This difference could be due to either the slower splicing kinetics of IVS9, or simply the presence of a *TOMM40* mRNA transcript (ENST592434) that retains the entire intron 9 in its mRNA structure.

After resolving the complication of pseudogene RNAs co-measurement, we demonstrated that the surrogate transcription level of *TOMM40* RNA is roughly 2.5-fold higher in AD compared to the control frontal lobe. Our results are opposite to prior published studies showing that *TOMM40* RNA is downregulated in AD blood [35,36,37,38]. Although this opposition could reflect different *TOMM40* regulatory pathways between blood and the CNS, it also suggested that the transcription levels of *TOMM40* pseudogene RNAs could vary across different tissues. Increased *TOMM40* RNA transcription in AD brains could be a consequence of prolonged mitochondrial dysfunction, which triggers a feedback response to upregulate structural proteins (e.g., TOM40) to compensate for the compromised mitochondrial function. The upregulated *TOMM40* RNA in AD brains has the same trend as the upregulation of *APOE* mRNA in AD brains compared to control [40]. The consistent upregulation of both *TOMM40* and *APOE* in AD brains makes the concept of co-regulation of these genes through the same topological associating domain even more appealing. Whether the upregulation of *TOMM40* RNA is truly associated with AD risk remains to be further validated with a larger sample size and across different brain regions. Nevertheless, this study provides a new molecular tool for measuring *TOMM40* RNA, making future expanded studies feasible.

Our genetic association analyses revealed no difference between *TOMM40* RNA transcription levels with either *APOE* rs429358 variants (C/ε4+ vs. C/ε4−) or *TOMM40* rs2075650 variants (G+ vs. G−), two SNPs that have been consistently linked to AD risk through GWAS studies. This lack of association suggested that these two genetic variants may not have a direct impact on the increased *TOMM40* RNA levels in AD. On the contrary, the SNPs rs10527523 and rs741780 showed significant allelic differences associated with *TOMM40* RNA transcription. Because these two SNPs are located between introns 6 and 8 of the *TOMM40* gene, this region of *TOMM40* might contain functional regulatory elements that influence the transcription of *TOMM40*. In the case of rs10524523, we observed a higher transcription level (lower ΔC_T_ value) of *TOMM40* RNA in the “S” variant when compared to the “VL” allele. This result is opposite to the study of Linnertz et al., who showed *TOMM40* mRNA levels were lower in “S” homozygotes compared with “VL” homozygotes in the AD brain [24]. Again, the inclusion or exclusion of *TOMM40* pseudogene RNAs transcription levels could account for these conflicting results.

Mitochondria are involved in several cellular functions and are essential for energy production; they are the main organelles that provide energy for brain cells. Indeed, neurons are particularly sensitive to changes in mitochondrial function [69], and mitochondrial injury can have severe consequences for neuronal function and survival [70]. We studied two mitochondrial function-related phenotypes (copy number and membrane potential) and their associations with *TOMM40* RNA transcription levels in human cell lines. MtDNA copy number is a measure of the number of mitochondrial genomes per cell and is a proxy for mitochondrial function [71,72,73]. Significant differences in this copy number have been reported across different brain regions, and these variations were more pronounced in patients affected by neurodegenerative disorders [74]. Studies have also shown that MtDNA levels were decreased by 30–50% in the frontal cortex of AD patients when compared to controls [75,76]. Mitochondrial membrane potential, which is used by ATP synthase to make ATP, serves as an intermediate form of energy storage for cells. Normally, cells maintain stable levels of mitochondrial membrane potential to carry out various cellular functions [77,78,79]. This membrane potential is altered due to physiological activity on a transient basis, but a prolonged alteration could compromise the viability of the cells and cause irreversible damage [80]. Our analyses showed that the increased *TOMM40* RNA levels are associated with a lower MtDNA copy number and a lower mitochondrial membrane potential, which together signified a decrease in mitochondrial function. This observation could be explained by a feedback response to restore mitochondria function via upregulation of *TOMM40* mRNA. As a translocase of outer mitochondrial membrane [9,10], TOM40 protein plays a role in importing proteins for the assembly of the mitochondrial inner membrane respiratory chain and mitochondrial matrix proteins involved in oxidative respiration. Increased *TOMM40* RNA transcription associated with AD could lead to changes in mitochondrial protein import, which might affect maintenance of mitochondrial membrane potential and overall mitochondrial function.

## 5. Conclusions

Here, we developed a novel assay to measure true *TOMM40* RNA transcription levels with high specificity and sensitivity, circumventing the unintentional co-measurement of *TOMM40* pseudogene RNAs. This assay enabled us to accurately investigate the RNA transcription profile of *TOMM40* associated with AD. The PMB work showed that *TOMM40* mRNA is upregulated in AD vs. control frontal lobe. The ex vivo cultured cell line work showed that upregulation of *TOMM40* RNA is likely associated with compromised mitochondrial function. Although future work using a larger sample size is needed to replicate these results, this work pioneers a valuable blueprint to assess *APOE*-independent effects of *TOMM40* in AD risk. Our findings define a new paradigm of *TOMM40* gene regulation and provide novel insight into the transcriptional pathway of *TOMM40*. This pathway involves not only the production of multiple *TOMM40* mRNA species, but also a pseudogene-imparted transcriptional program. Many epidemiology studies strongly support the idea that *TOMM40* contributes to healthy aging. Because age is the most important known risk factor for AD, it raises an interesting question: could the incidence of AD be a byproduct of a compromised longevity pathway that is carefully guarded via *TOMM40*-imparted mitochondrial function?

## Figures and Tables

**Figure 1 genes-12-00871-f001:**
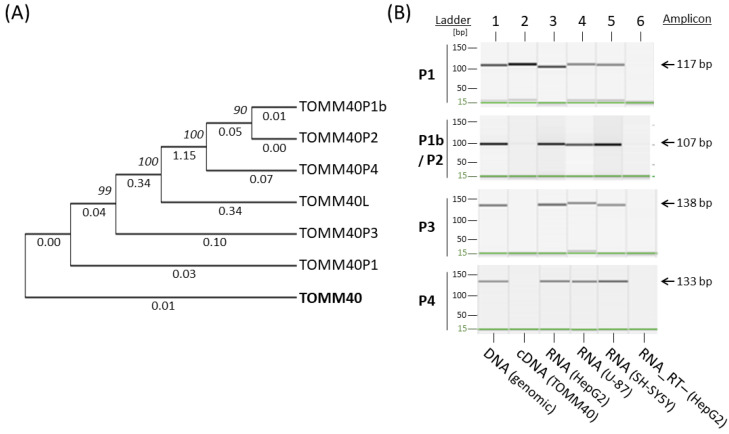
Phylogenetic tree evolution and end-point RT-PCR amplification of *TOMM40* pseudogenes. (**A**) Phylogenetic tree structure of *TOMM40*-related genes and pseudogenes. Numbers above the lines are bootstrap value (or node), and numbers below the lines are substitution rate. (**B**) Capillary gel electrophoresis images of RT-PCR amplified pseudogenes from total RNA of three human cell lines (HepG2, U-87, and SH-SY5Y). Amplicons for pseudogenes *P1*, *P1b*, *P2*, *P3*, and *P4* are shown. Genomic DNA (lane 1) served as a positive control, *TOMM40* cDNA (lane 2) served as a reference for cross-amplification, and RT- RNA (lane 6) served as a negative control.

**Figure 2 genes-12-00871-f002:**
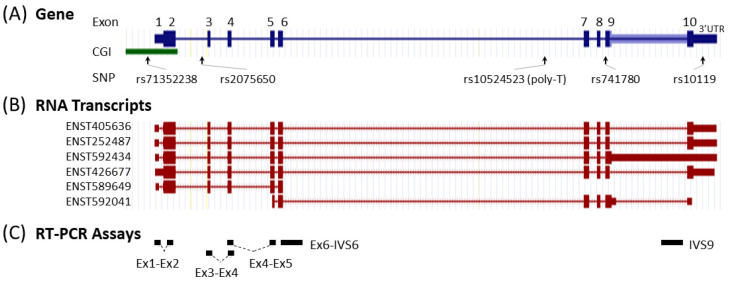
Map of the *TOMM40* gene, RNA transcripts, and RT-PCR assays. (**A**) Structure and position of the exons, CpG island (CGI), and genetic variants (SNP). (**B**) Structure of the six mRNA transcripts defined by the Ensembl database. (**C**) Targets of the RT-PCR assays used in this study.

**Figure 3 genes-12-00871-f003:**
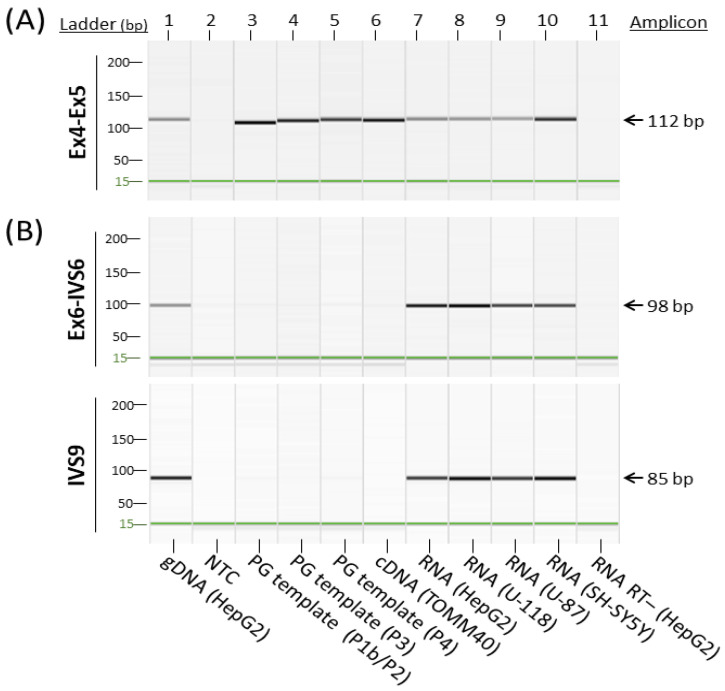
Comparison between a conventional cDNA assay and primary transcript-targeted cDNA assays. Capillary gel electrophoresis images of the RT-PCR amplified *TOMM40* pseudogene amplicons. (**A**) *TOMM40* primer set spans Ex4 and Ex5 that mimics a commercial TaqMan gene expression assay (Thermo Fisher, Waltham, MA, USA, Hs01587378_mH). (**B**) *TOMM40* primer sets targeting primary transcripts of exon 6 to intron 6 (EX6-IVS6) and intron 9 (IVS9). Genomic DNA (lane 1) served as a positive control; no-template control (NTC, lane 2) and RT- RNA (lane 11) served as a negative control.

**Figure 4 genes-12-00871-f004:**
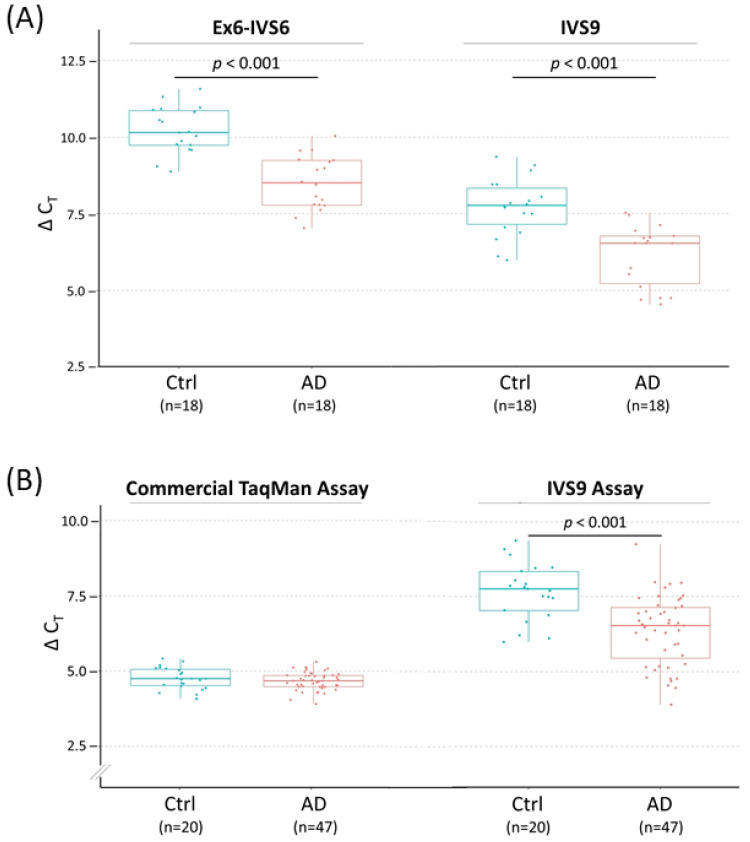
*TOMM40* RNA transcript levels in human PMB tissues by RT-qPCR quantification. Transcription levels of *TOMM40* RNAs are plotted as values of ΔC_T_ (mean of C_T_ triplicates (target)—mean of *ACTB* C_T_ triplicates) and compared between control (Ctrl) (blue) and AD (red) frontal lobes. In this setting, smaller ΔC_T_ values indicate higher RNA levels. (**A**) Comparison of the two *TOMM40* primary transcript-targeted assays (Ex6-IVS6 vs. IVS9) as the pilot study. (**B**) Comparison of a commercial TaqMan cDNA assay (Thermo Fisher, assay Hs01587378_mH) and the primary transcript-targeted TaqMan IVS9 assay with expanded samples. Numbers in parentheses denote sample size. The *t*-test *p* values are shown where significant differences between Ctrl and AD were detected. Boxplot shows quartiles and median.

**Figure 5 genes-12-00871-f005:**
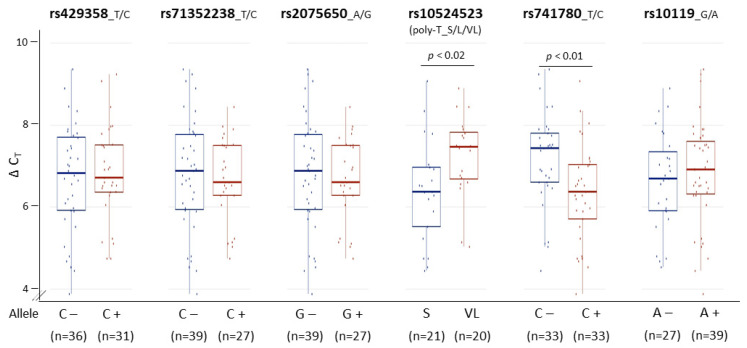
Associations of PMB *TOMM40* RNA levels with genetic variants. Transcription levels of PMB *TOMM40* RNA were measured by the IVS9 assay and compared across six genetic variants. The *t*-test *p* values are shown where significant associations with *TOMM40* intron 6 poly-T SNP rs10524523 (*p* < 0.02) and intron 8 SNP rs741780 (*p* < 0.01) were detected. For rs10524523, the S group includes S/S homozygotes and S/L heterozygotes, and the VL group includes VL/VL homozygotes and VL/L heterozygotes. Numbers in parentheses denote sample size. Boxplot shows quartiles and median.

**Figure 6 genes-12-00871-f006:**
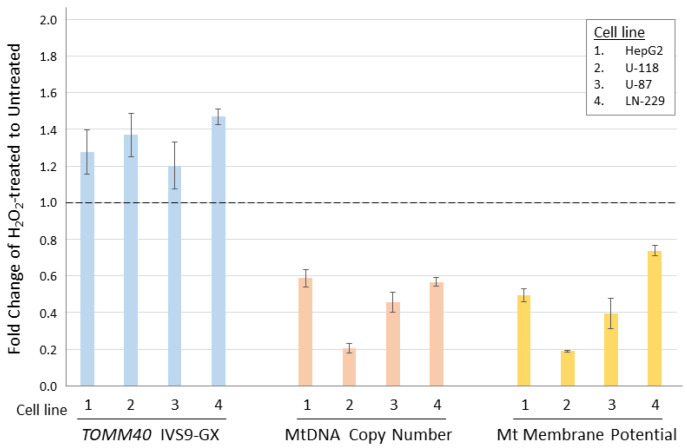
Comparison of *TOMM40* RNA levels with mitochondrial functions in human cell lines. *TOMM40* RNA levels were measured by the IVS9 assay and mitochondrial function assays including MtDNA copy number and Mt membrane potential were performed. The fold change of H_2_O_2_-treated cells was compared to the untreated cells (set as baseline of 1.0). Graph shows the relationship between the three measurements in response to the oxidative stress. Standard deviation error bars are shown.

**Table 1 genes-12-00871-t001:** Demographics of the PMB study samples.

Subjects	AD	Control
Sample number—*n*	47	20
Gender—*n* female (% female)	27 (57.4)	11 (55.0)
APOE ε4+—*n* (%)	29 (61.7)	3 (15.0)
Age at death—mean (SD)	87.9 (5.9)	88.3 (8.5)
Age at onset—mean (SD)	79.0 (8.0)	N/A
Disease duration—mean years (SD)	9.0 (4.4)	N/A
Postmortem interval—mean hours (SD)	5.0 (2.0)	4.9 (2.3)
CERAD Score		
Absent	0	7
Sparse	0	7
Moderate	11	4
Frequent	36	2
Braak Stage		
I	0	6
II	0	11
III	0	3
IV	0	0
V	15	0
VI	32	0

SD: standard deviation.

**Table 2 genes-12-00871-t002:** Genomic locations and RNA transcripts of *TOMM40*-related genes and pseudogenes.

Gene/Pseudogene	Genomic Sequence (hg38)		RNA Sequence (RefSeq)		
	Coordinate	Span (bp)	Accession #	Size (nt)	BLASTn **
*TOMM40*	chr19: 44891220-44903689	12,470	NM_001128916	1676	-
*TOMM40P1*	chr14: 19266948-19268660	1713	NG_022836	1713 *	95.70%
*TOMM40P1b*	chr14: 19131227-19133057	1831	N/A	1831 *	95.55%
*TOMM40P2*	chr22: 15853581-15855410	1830	NG_022885	1830 *	95.88%
*TOMM40P3*	chr5: 3501872-33503327	1456	NG_021878	1456 *	87.26%
*TOMM40P4*	chr2: 31723017-131724478	1462	NG_023610	1462 *	95.41%
*TOMM40L*	chr1: 161226060-161230746	4687	NM_032174	2790	70.17%

*: Putative RNA size estimated from corresponding genomic locus. **: % identify when compared to *TOMM40* mRNA.

## Data Availability

All relevant data are within the manuscript and its Appendix A.

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
