# Peer review of "TOMM40 RNA Transcription in Alzheimer’s Disease Brain and Its Implication in Mitochondrial Dysfunction"

_genes, 2021, doi:10.3390/genes12060871_

Round 1

Reviewer 1 Report

It is a well-written manuscript. The reviewer only has one minor comment.

Line 566-568. TOMM40 is an outer membrane protein. Mitochondrial mmebrane potential is dependent on the integrity of mitochondrial inner membrane. Could authors give some explanation how alteration of TOMM40 leads to depolarization of mitochondria membrane potential?

Author Response

  1. Line 566-568. TOMM40 is an outer membrane protein. Mitochondrial membrane potential is dependent on the integrity of mitochondrial inner membrane. Could authors give some explanation how alteration of TOMM40 leads to depolarization of mitochondria membrane potential?"

Response: We add these following sentences to the end of Discussion section (lines 605-611): “TOMM40 encodes a translocase of outer mitochondrial membrane, and this channel-forming protein is essential for import of proteins into mitochondria (with two references cited). TOM40 protein plays a role in importing proteins for assembly of the mitochondrial inner membrane respiratory chain and mitochondrial matrix proteins involved in oxidative respiration. Increased TOMM40 RNA transcription associated with AD could lead to changes in mitochondrial protein import, which might affect maintenance of mitochondrial membrane potential and overall mitochondrial function.”

Reviewer 2 Report

The paper “TOMM40 RNA expression in Alzheimer’s disease brain and its implication in mitochondrial dysfunction” studies the expression of TOMM40 RNA in post-mortem brains of Alzheimer’s disease patients and its relation with mitochondrial dysfunction using cell lines. Additionally, the authors developed a novel assay to measure the TOMM40 RNA expression, avoiding the co-measurement of TOMM40 pseudogenes RNAs.

The paper is very interesting, with useful information about the correct measurement of TOMM40 RNA expression. The authors explained why they recommend the quantification of the primary transcript of TOMM40 and not the spliced RNA, presenting the limitations of the current available methods. Also, the authors discuss the importance to study the role of TOMM40 pseudogenes in the control of TOM40 levels.

The paper is well-written and well-structured. I have only a few recommendations:  

1) 500µM Hydrogen peroxide was used to induce oxidative stress in cell lines. However, the authors did not explain the selection of this concentration of hydrogen peroxide. Did the authors tested several concentrations of hydrogen peroxide or did they follow the available literature? This question should be clarified in the manuscript.

2) The manuscript lacks a section of statistical analysis. What was the statistical test used to analyze the graphs in figure 4 and in figure 5? Are there statistical differences between the data of mtDNA copy number or of the mitochondrial membrane potential?

3) Please clarify and support the sentence: “Although expression levels are not 508 the same between primary transcript and spliced mRNA”.

4) In the discussion: “MtDNA copy number is a measure of the number of mitochondrial genomes per cell and is a proxy for mitochondrial function.” What is the bibliographic reference that supports the second part of this sentence? MtDNA is commonly accepted as an indicator of mitochondrial mass and it is not directly correlated to the mitochondrial function.

5) I would expect to read in the discussion more information regarding TOM40 and mtDNA and mitochondrial membrane potential. For example, why TOM40 levels are increased in Alzheimer’s disease brains? Is it possible that there is an increase of TOM40 levels to favor the mitochondrial import in case of reduced mitochondrial membrane potential?

6) The legend of the figure 1B lacks information about what is the P1, P1b/P2, P3 and P4. Thus, this legend is not self-explanatory.

Author Response

Response to Reviewer 2 Comments

  1. 500µM Hydrogen peroxide was used to induce oxidative stress in cell lines. However, the authors did not explain the selection of this concentration of hydrogen peroxide. Did the authors tested several concentrations of hydrogen peroxide or did they follow the available literature? This question should be clarified in the manuscript.

Response: In response to this comment, we have added these following sentences to the Materials and Methods section (lines 195-199). “ We searched the literature for the effects of hydrogen peroxide on mitochondrial function. Based on previously published conditions, we tested multiple concentrations (100 mM, 200 mM, 250 mM, 500 mM and 1 mM) of hydrogen peroxide in the cell lines and selected 500 uM as an optimal concentration which maintained good cell viability and had markable effects on mitochondrial function.”

  1. The manuscript lacks a section of statistical analysis. What was the statistical test used to analyze the graphs in figure 4 and in figure 5? Are there statistical differences between the data of mtDNA copy number or of the mitochondrial membrane potential?

Response: We have integrated a Statistical Analyses section to the Materials and Methods (lines 238-245). Our response to the second question is as follows: Although hydrogen peroxide treatment conditions were the same in both MtDNA copy number and membrane potential experiments, these two experiments could not be performed in the same cell cultures. Therefore, a direct comparison between the two assays was not feasible.  

  1. Please clarify and support the sentence: “Although expression levels are not 508 the same between primary transcript and spliced mRNA”.

Response: We have modified this sentence as follows: “It has been shown that the splicing efficiency of pre-mRNA varies greatly across genes.” together with two new references cited. Due to the splicing process expression levels are often not the same between primary transcript and spliced mRNA. However, the level of primary transcript can provide a surrogate measurement for mRNA”. (lines 538-542).

  1. In the discussion: “MtDNA copy number is a measure of the number of mitochondrial genomes per cell and is a proxy for mitochondrial function.” What is the bibliographic reference that supports the second part of this sentence? MtDNA is commonly accepted as an indicator of mitochondrial mass and it is not directly correlated to the mitochondrial function.

Response: We have added three new references to support the concept that MtDNA copy number as a proxy for Mt function (Ref. 71-73, line 592).

  1. I would expect to read in the discussion more information regarding TOM40 and mtDNA and mitochondrial membrane potential. For example, why TOM40 levels are increased in Alzheimer’s disease brains? Is it possible that there is an increase of TOM40 levels to favor the mitochondrial import in case of reduced mitochondrial membrane potential?

Response: We appreciate this Reviewer’s suggestion. We have incorporated the following to the Discussion section. “Increased TOMM40 transcription levels in AD brain, may be upregulating TOM40 protein to favor mitochondrial import. This could be a natural mitochondrial response to restore the compromised mitochondrial membrane potential (lines 605-608).

  1. The legend of the figure 1B lacks information about what is the P1, P1b/P2, P3 and P4. Thus, this legend is not self-explanatory.

Response: We have modified the Figure 1 legend to include the following:  “Amplicons for pseudogenes P1, P1b, P2, P3 and P4 are shown.” (lines 274-275).

Reviewer 3 Report

The manuscript entitled “TOMM40 RNA Expression in Alzheimer’s Disease Brain and its Implication in Mitochondrial Dysfunction” (genes-1241487) by Lee et al. analyzes the transcription of the TOMM40 gene as a potential marker for Alzheimer’s disease.

The TOMM40 gene, which encodes for the TOM40 protein, has been identified in genome-wide association studies for its linkage to Alzheimer’s disease (AD). However, its proximity to the APOE gene locus, which shows an even stronger linkage to AD, has long obscured the linkage by TOMM40 gene. Increasing evidence suggests that TOMM40 may affect the pathogenesis of AD in its own way, since it encodes for a translocase of the outer mitochondrial membrane complex (TOM40) and could affect the function of neuronal mitochondria.

Detection of the TOMM40 gene expression is often done with commercially available detection kits that measure transcript levels in cells or tissues. However, the presence of multiple pseudogenes of TOMM40 in the human genome complicates these analyzes. In the current manuscript the authors use both commercially available detection kits as well as a variety of primer pairs to assay the transcript levels of TOMM40 and its previously unrecognized pseudogenes. Transcripts from the relatively well-preserved pseudogenes were found to interfere with the detection of the TOMM40 mRNA transcripts and an alternative set of primers were developed to accurately measure the transcripts from TOMM40 only.

Given the detrimental effects erroneous measurements of the TOMM40 transcript levels may cause in AD research, this study is both timely and important. However, in its current form the manuscript has a number of weaknesses that need to be addressed:

  • Throughout the manuscript the authors write about “TOMM40 RNA expression” or “TOMM40 expression” when really TOMM40 transcription is implied. It would benefit the clarity of the manuscript to employ a more strict usage of scientific terms and exchange most uses of the term “expression” with the more precise term “transcription”, including the title of the study.
  • As the detection of TOMM40 transcripts and of it’s pseudogenes is a central part of this study, the list of primers that were used for the different detection attempts is a crucial part of the manuscript. Therefore, the primer list in Table S1 should be made part of the main manuscript. Moreover, the primers themselves should be listed in a non-proportional font (e.g. Courier) so that the primer sequences can be aligned for easy comparison.
  • Similarly, the graphs in Figure S3 are of great interest and should be included in Figure 4 of the main manuscript. Since no statistical comparison is provided, it is not clear if the differences in Figure S3 are statistically significant or not. Such an analysis should be included.
  • In contrast, the data in Figure 6 are not well integrated with the rest of the manuscript as they only analyzed human cell lines. Moreover, they do not provide insights that are central to the focus of the study. Therefore, these data could be relegated to the supplemental section or outright removed.

Author Response

Response to Reviewer 3 Comments

  1. Throughout the manuscript the authors write about “TOMM40 RNA expression” or “TOMM40 expression” when really TOMM40 transcriptionis implied. It would benefit the clarity of the manuscript to employ a more strict usage of scientific terms and exchange most uses of the term “expression” with the more precise term “transcription”, including the title of the study.

Response: We have changed “TOMM40 RNA expression” to “TOMM40 transcription level” throughout the manuscript.

  1. As the detection of TOMM40 transcripts and of it’s pseudogenes is a central part of this study, the list of primers that were used for the different detection attempts is a crucial part of the manuscript. Therefore, the primer list in Table S1 should be made part of the main manuscript. Moreover, the primers themselves should be listed in a non-proportional font (e.g. Courier) so that the primer sequences can be aligned for easy comparison.

Response:  We have changed the font of Table S1 as suggested by the Reviewer; however, we did not move the Table S1 into the main manuscript for the following two reasons: 1) This is a large table that contains a lot of information with small font size. If moved into the main manuscript, it must comply with the font size requirement and will generate multiple line breaks making it hard to read. 2) Majority of the readers may not have an interest of checking into these primer sequences; therefore, keeping the main manuscript concise should be beneficial to most readers.

  1. Similarly, the graphs in Figure S3 are of great interest and should be included in Figure 4 of the main manuscript. Since no statistical comparison is provided, it is not clear if the differences in Figure S3 are statistically significant or not. Such an analysis should be included.

Response: We have merged Figure S3 with Figure 4 to generate a new Figure 4 and new legend  (Page 11, lines 410-418).

  1. In contrast, the data in Figure 6 are not well integrated with the rest of the manuscript as they only analyzed human cell lines. Moreover, they do not provide insights that are central to the focus of the study. Therefore, these data could be relegated to the supplemental section or outright removed.

Response: We appreciate this Reviewer’s suggestion, but we feel strongly that Figure 6 should be kept in the main manuscript for the following reason. Outside of genetic associations, this figure provides an important implication of the overall results to a potential phenotype of mitochondrial function, which will be of great interest for the readers.